# Risk of New-Onset Diabetes Mellitus Associated with Antirheumatic Drugs in Patients with Rheumatoid Arthritis: A Nationwide Population Study

**DOI:** 10.3390/jcm11082109

**Published:** 2022-04-10

**Authors:** So Hye Nam, Minju Kim, Ye-Jee Kim, Soo Min Ahn, Seockchan Hong, Chang-Keun Lee, Bin Yoo, Ji-Seon Oh, Yong-Gil Kim

**Affiliations:** 1Division of Rheumatology, Department of Internal Medicine, Uijeongbu Eulji Medical Center, Eulji University School of Medicine, Uijeongbu 11759, Korea; namsohye@hanmail.net; 2Division of Rheumatology, Department of Internal Medicine, Asan Medical Center, University of Ulsan College of Medicine, Seoul 05505, Korea; lisianthus@amc.seoul.kr (S.M.A.); medivineluke@gmail.com (S.H.); cklee@amc.seoul.kr (C.-K.L.); byoo@amc.seoul.kr (B.Y.); 3Department of Clinical Epidemiology and Biostatistics, Asan Medical Center, Seoul 05505, Korea; minjukim@amc.seoul.kr (M.K.); kimyejee@amc.seoul.kr (Y.-J.K.); 4Department of Information Medicine, Big Data Research Center, Asan Medical Center, Seoul 05505, Korea

**Keywords:** rheumatoid arthritis, diabetes mellitus, disease-modifying antirheumatic drugs

## Abstract

Background: This study aimed to investigate the effect of disease-modifying antirheumatic drugs (DMARDs) on diabetes mellitus (DM) development in rheumatoid arthritis (RA). Methods: This nested case–control study with a cohort of 69,779 DM-naïve adult patients with RA was conducted from 2011 to 2019 in South Korea. Cases with incident DM were identified and individually matched to randomly selected controls (1:4). DMARDs use was measured for 1 year before the index date and stratified by exposure duration. The association of each DMARD use with DM risk was estimated using conditional logistic regression adjusted for comorbidities and concomitant drug use. Results: Of the patients, 5.4% were newly diagnosed with DM. The use of statins and a higher cumulative dose of corticosteroids were associated with an increased DM risk. In a multivariable-adjusted analysis, cumulative duration of exposure (CDE) >270 days/year, hydroxychloroquine (HCQ; adjusted odds ratio [aOR], 0.76) and methotrexate (MTX; aOR, 0.81) were associated with a significant decrease in DM risk, and tacrolimus (TAC; aOR, 1.27) was associated with an increased risk. Conclusions: Long-term use of HCQ and MTX (>270 days/year) was associated with a reduction in DM incidence as opposed to TAC.

## 1. Introduction

Rheumatoid arthritis (RA) is a chronic systemic inflammatory disease that mainly affects the joints, resulting in loss of function and lowering quality of life. Aside from the loss of joint function, RA patients suffer from combined comorbidities, specifically cardiovascular disease (CVD), which is the most common cause of mortality and morbidity in patients with RA [1,2,3,4,5]. As is well known, diabetes mellitus (DM) is a traditional risk factor for CVD, and a higher cardiovascular risk is observed in RA patients with DM [6,7,8]. The prevalence of DM is reported to be 6–20% in patients with RA, and a recent study revealed that RA is associated with a 23% increased DM risk [9,10]. The reasons for this high prevalence of DM in RA suggests that several predisposing factors including obesity, dyslipidemia, and corticosteroids can contribute to the development of DM [11,12]. In addition, systemic inflammation in RA induced by proinflammatory cytokines such as tumor necrosis factor (TNF)-α and interleukin (IL)-6 is also known to be linked with insulin resistance by blocking the function of insulin at the receptor level [13,14]. Thus, several studies exist on the association between RA treatment and DM, attempting to find a treatment regimen that can reduce DM incidence [11,15,16,17,18,19]. Hydroxychloroquine (HCQ) has shown favorable incident DM outcomes in large epidemiological studies [15,18]. However, inconsistent data exist on disease-modifying antirheumatic drugs (DMARDs), except for HCQ [11,15,16,17,18]. Furthermore, assessing the effects of DMARDs on incident DM is difficult because most studies are heterogeneous in design. 

Therefore, the association of individual DMARDs with incident DM was investigated here, and whether the sustainability of DMARDs affects DM development in patients with RA was examined. 

## 2. Materials and Methods

### 2.1. Data Source

Korean Health Insurance Review and Assessment Service (HIRA) claims data from 2011 to 2019 were used. HIRA includes all health-related information from around 50,000,000 individuals across the entire Korean population covered by the National Health Insurance program [20]. This information includes patient demographics, medical diagnoses, hospital referrals, and drug prescriptions. Disease diagnoses in the data were performed using the International Classification of Diseases, Tenth Revision (ICD-10). 

### 2.2. Study Population

A nested case–control study of a nationwide RA population was conducted with the HIRA data. Patients with newly diagnosed RA (ICD-10 code, M05) from 1 October 2011 to 30 September 2019 were selected for the RA cohort. For diagnostic validation, we included RA patients who visited the outpatient clinic at least twice, had a prescription of DMARDs, and who were listed in the Korean rare and intractable diseases (RID) system. In the Korean RID system, RA is diagnosed based on the 2010 American College of Rheumatology/European League Against Rheumatology classification criteria for RA and on a positive test for rheumatoid factor or anticitrullinated peptide antibodies [21]. The selection of the study population is outlined in Figure 1. Patients who received a diagnosis of DM or were prescribed any antidiabetic medication the year before base-cohort entry were excluded. Patients were also excluded if they had any of the following characteristics during the 1 year prior to the baseline, because the treatment of underlying comorbidities was expected to make it more difficult to assess the impacts of RA treatment: a diagnosis of (1) other rheumatic diseases including systemic lupus erythematosus and mixed connective tissue disease, (2) interstitial lung disease, (3) transplantation, (4) cancer, (5) human immunodeficiency virus infection, and/or (6) end-stage renal disease requiring dialysis (as detailed in Appendix A).

### 2.3. Case and Control Definition

Cases were defined as having their first diagnosis of diabetes from October 2012 to September 2019. The definition of diabetes was a newly recorded diagnosis of DM (ICD-10, disease code E10-14) combined with a prescription for antidiabetic medication (all insulin preparations as well as oral hypoglycemic agents). Each case was matched to four controls by sex (male/female), age at RA diagnosis (±1 year), and year of RA diagnosis (duration of the disease until DM diagnosis). The controls were defined among the current cohort population. The index date was defined as the first date of diabetes diagnosis, and the index date for controls was their matched case’s index date.

### 2.4. Exposure Measurement

Exposure to antirheumatic agents was identified via the prescription records. HCQ, methotrexate (MTX), leflunomide (LEF), sulfasalazine (SSZ), and tacrolimus (TAC) were classified into conventional DMARDs (cDMARDs). The use of cDMARDs was defined as having filled at least two prescriptions for the drugs within 365 days before the index date, and non-user was defined as having no record using DMARDs. The association between cDMARD exposure level and the risk of diabetes incidence was investigated by the cumulative duration of exposure (CDE). Moreover, the CDE was defined as the sum of the exposure duration for each drug and classified as 0 (nonusers), 1–90, 91–180, 181–270, and 271–365 days.

### 2.5. Confounding Variables

All covariates which are important diabetes determinants were assessed using data from the 365 days before the index date. Data including age, gender, comorbid medical conditions (e.g., asthma, chronic obstructive pulmonary disease [COPD], CVD, and chronic kidney disease), and use of medications were collected for each patient (Appendix A). The Charlson Comorbidity Index (CCI) was calculated to assess underlying comorbidities except for connective tissue disease [22]. In addition, prescriptions of statins, corticosteroids, and other DMARDs (e.g., biologic DMARDs) and targeted synthetic DMARDs (tsDMARDs) were examined and also included as covariables. Glucocorticoid exposure was categorized following the cumulative equivalent dosage of oral prednisolone, considering the well-known association between corticosteroid use and diabetes development [23].

### 2.6. Statistical Analysis 

Categorical and continuous variables were presented as numbers with percentages and median with interquartile range, respectively. The balance in the distribution of baseline characteristics between DM patients and matched controls was quantified using the standardized difference of the mean (SMD). A SMD < 0.1 was regarded as a fair balance of confounders between the matched group. The impact of each DMARD individually on DM risk compared with its non-user using conditional logistic regression models was determined. The risk of new-onset DM with adjustment for CCI, cumulative steroid use, statin use, and use of other DMARDs was also assessed.

Results were reported using adjusted odds ratios (ORs) and their 95% confidence intervals (CIs). *p* < 0.05 was considered statistically significant. All statistical analyses were performed using SAS Enterprise Guide software version 7.1 (SAS Institute, Inc., Cary, NC, USA).

## 3. Results

Among the 69,779 RA patients who met the inclusion and exclusion criteria, 3772 (5.4%) cases newly diagnosed with DM and 14,830 controls matched by gender, age at RA diagnosis, and year of RA diagnosis were identified (Figure 1). The cases had a mean age of 62.3 ± 10.9 years at the index date, and 77.6% of them were women (Table 1). The mean duration from RA diagnosis to DM incidence was 2.6 ± 2.1 (median, 2.1; range, 0.0–8.0) years. The cases showed a higher prevalence of comorbidities and comedications, including essential hypertension, hypercholesterolemia, ischemic heart disease, asthma, COPD, CCI, use of statins, corticosteroids, and cDMARDs (Table 1). 

Table 2 shows the ORs of incident DM for the comedications within a year of indexing. Concurrent use of statins significantly increased the odds of incident DM (unadjusted OR = 2.37, 95% CI = 2.19–2.56; adjusted OR [aOR] = 2.17, 95% CI = 2.00–2.35). Corticosteroid use showed higher odds with increasing cumulative doses, with an aOR of 3.20 (2.82–3.62) in the highest quintile (335.5–6681.8 mg) compared with the lowest quintile (0–7 mg). Based on these results, analysis of the association between cDMARDs and DM incidence was performed by adjusting for co-medications such as statins and corticosteroids.

Table 3 shows the association between the use of cDMARDs more than twice within a year and the risk of incident DM. The use of HCQ and MTX were associated with decreased aOR of incident DM; otherwise, LEF and TAC were associated with increased aOR. To investigate the potential impact of past exposure and exposure duration of DMARDs, the dose–response relationship for the risk of developing DM according to the CDE for each cDMARD was evaluated. Figure 2 shows that CDE ≤ 90 days/year was associated with an increased risk of developing DM compared with non-exposure for all individual cDMARDs. In contrast, aORs tended to significantly decrease with increasing CDE for all cDMARDs, and HCQ (aOR = 0.76, 95% CI = 0.69–0.84) and MTX (aOR = 0.81, 95% CI = 0.74–0.89) were associated with a significant reduction in DM risk at CDE >270 days/year. However, TAC still showed a significantly increased DM risk even at CDE >270 days/year (Figure 2; Appendix A).

## 4. Discussion

The present study showed the dose–response relationship between cDMARDs and incident DM by CDE. The use of MTX and HCQ more than 270 days/year was associated with a lower incidence of new-onset DM in multivariable analyses adjusted for comorbidities and concomitant medications. 

DM can be developed with high inflammatory status and controlling the disease activity of RA while simultaneously minimizing the use of corticosteroids is important. In a previous cohort study on the association of corticosteroid use and incident DM in RA patients, DM risk was increased with higher dosage and longer exposure, and current use of low-dose corticosteroids (prednisolone 5 mg/day) was also associated with DM [23]. The data from the current study consistently showed that DM risk increased as much as the cumulative dose of corticosteroids, despite the possibility that the results were underestimated because we did not include intramuscular or intravenous corticosteroid injections. Several studies have documented the relationship between statin use and incident DM [18,24]. However, as presented in Table 1, the proportion of patients with dyslipidemia was higher in the case group. Considering that DM is associated with metabolic syndrome, which includes obesity, dyslipidemia, and hypertension, it is important to manage these factors to prevent DM development [25]. Consequently, multivariable analyses were conducted by adjusting for various risk factors including comorbidities, corticosteroids, statins, and other DMARDs to evaluate the effect of individual DMARDs on DM. Among the DMARDs, HCQ is known as a representative DMARD that has a protective effect on DM by improving insulin sensitivity and pancreatic β-cell function [26]. In the US cohort study, HCQ and abatacept were associated with decreased risk of incident DM, and, in particular, the longer HCQ was treated (>4 years), the greater the reduction in DM risk was observed compared with non-HCQ users [18]. MTX, as a key RA treatment, has been focused on reducing atherosclerosis and incident DM by controlling inflammation. Although studies on MTX’s effect on atherosclerosis have been inconclusive, most DM studies have shown a decreased risk of incident DM [15,16,27,28,29]. Other cDMARDs (e.g., SSZ and LEF) have been suggested to have antihyperglycemic effects. However, clinical data to support this are lacking [30,31].

In present study, among cDMARDs, HCQ and MTX in particular showed significantly reduced DM risk at CDE > 270 days/year. Moreover, SSZ and LEF were associated with an increase in DM risk at CDE ≤ 90 days/year, but no longer showed a significant increase in DM risk at CDE > 270 days/year. These findings may suggest that the inadequate treatment of RA or a high disease activity status requiring corticosteroids use, rather than cDMARD use itself, was associated with an increased DM risk, and that continued cDMARD use was associated with a reduced DM risk. In addition, a recent study reported that an increased risk of incident DM was associated with higher disease activity and elevated inflammatory cytokine/chemokine levels (IL-6, IL-1, macrophage-derived cytokine/chemokines) in RA patients [14]. Therefore, efforts to achieve low disease activity in RA and improve patient compliance are needed in clinical practice to reduce the prevalence of incident DM in patients with RA. However, TAC was associated with an increase in DM risk at CDE ≤ 90 and CDE > 270 days/year, consistent with the well-known TAC diabetogenicity [32]. Although TAC is not commonly used for RA treatment, there are patients who have an inadequate response to csDMARDs but are difficult to treat with biologic or tsDMARDs owing to socioeconomic status or underlying comorbidities [33,34,35]. In this case, physicians are recommended to monitor their blood glucose index (e.g., hemoglobin A1c) during treatment.

The present study has some limitations. First, a restriction of access to individuals’ information (e.g., family history of DM, body mass index, smoking, disease activity, and exact compliance to drugs) and the absence of laboratory results including actual blood glucose values were noted because the study design was retrospective and performed with claims data. Second, cumulative drug exposure only 1 year before DM diagnosis was calculated, which may be insufficient to assess the impact on DM. However, considering that the median duration between RA diagnosis and incident DM was about 2 years, the observation period was not too short to assess the effect of treatment sustainability on incident DM. Third, biologic and tsDMARDs were not evaluated because of the small population. As increased TNF-α and IL-6 are associated with the pathogenesis of insulin resistance, biologic DMARDs (e.g., TNF inhibitors and tocilizumab) were expected to decrease DM risk by increasing insulin sensitivity [15,19]. In meta-analyses, TNF inhibitors use was associated with a significant decrease in diabetes risk [36]. However, the data on other biologic DMARDs including tocilizumab, abatacept, and tsDMARDs is still lacking [11,18,36]. Thus, further studies are needed as the use of biologic and tsDMARDs expands.

## 5. Conclusions

This study demonstrated a decreased risk of incident DM in RA patients treated with HCQ and MTX as opposed to TAC and corticosteroids, which increased the risk of incident DM. This protective effect of cDMARDs on the development of incident DM was more apparent with a longer treatment duration (>270 days/year). Although further studies are required to identify the effect of biologic and tsDMARDs, the findings of the current study emphasized that the use of HCQ and MTX are associated with a reduced risk of incident DM.

## Figures and Tables

**Figure 1 jcm-11-02109-f001:**
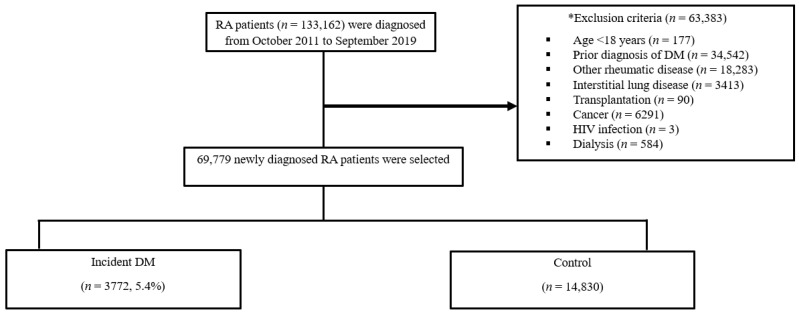
Flow diagram showing the selection illustration. RA: rheumatoid arthritis, DM: diabetes mellitus, HIV: human immunodeficiency virus * Other rheumatic disease were systemic lupus erythematosus and mixed connective tissue disease, Sjogren’s syndrome, and inflammatory myositis.

**Figure 2 jcm-11-02109-f002:**
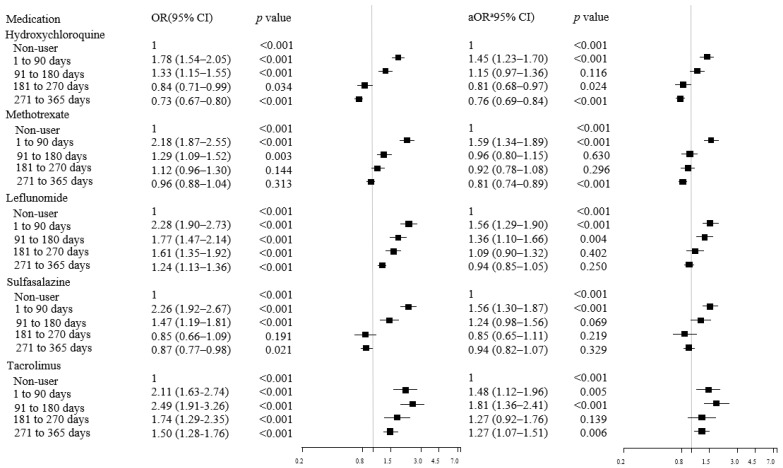
Risk of incident DM according to CDE (days/year) of each cDMARDs. DM: diabetes mellitus, CDE: cumulative duration of exposure, cDMARDs: conventional disease-modifying anti-rheumatic drugs, OR: odds ratio, aOR: adjusted odds ratio, CI: confidence interval, CCI: Charlson Comorbidity Index. ^a^ Adjusted for CCI (ordinal), use of steroid (cumulative prescribed dose quintiles), use of statin (yes/no), and use of other DMARDs.

**Table 1 jcm-11-02109-t001:** Clinical characteristics of cases and controls.

Characteristic	Case ^a^ (*n* = 3772)	Control ^a^ (*n* = 14,830)	SMD	*p* Value
Sex, female no. (%)	2928 (77.6)	11,526 (77.7)	0.00	NE
Age at RA diagnosis, mean ± SD, years	60.2 ± 11.0	59.9 ± 10.9	0.02	NE
Age at index ^b^ mean ± SD, years	62.3 ± 10.9	62.1 ± 10.9	0.02	NE
Comorbidities, no. (%)				
Essential hypertension	2012 (53.3)	5027 (33.9)	0.40	<0.001
Hypercholesterolemia	479 (12.7)	1210 (8.2)	0.15	<0.001
Ischemic heart disease	347 (9.2)	619 (4.2)	0.20	<0.001
Carotid stenosis	28 (0.7)	63 (0.4)	0.04	0.014
Asthma	508 (13.5)	1198 (8.1)	0.17	<0.001
COPD	121 (3.2)	253 (1.7)	0.10	<0.001
Ischemic stroke	166 (4.4)	414 (2.8)	0.09	<0.001
Chronic kidney disease	42 (1.1)	88 (0.6)	0.06	0.001
CCI score ^c^, mean ± SD	1.0 ± 1.1	0.7 ± 0.9	0.34	<0.001
0	1425 (37.8)	7740 (52.2)		
1–2	1983 (52.6)	6443 (43.4)		
3–4	325 (8.6)	591 (4.0)		
≥5	39 (1.0)	56 (0.4)		
Medications (ever use within a year), *n* (%)				
Statins	1466 (38.9)	3224 (21.7)	0.379	<0.001
Corticosteroids	1461 (38.7)	3211 (21.6)	0.379	<0.001
cDMARDs ^d^	3515 (93.2)	13,401 (90.4)	0.103	<0.001
Any TNFi	199 (5.3)	971 (6.5)	0.054	0.004
Non-TNFi	59 (1.6)	203 (1.4)	0.016	0.406
tsDMARDs	13 (0.3)	25 (0.2)	0.035	0.052

SMD: standardized mean difference, RA: Rheumatoid arthritis, SD: standard deviation, COPD: chronic obstructive pulmonary disease, CCI: Charlson comorbidity index, cDMARDs: conventional disease-modifying anti-rheumatic drugs, TNFi: tumor necrosis factor inhibitor, tsDMARDs: targeted synthetic disease-modifying anti-rheumatic drugs. a Data are given as number (percentage). b Age at index was defined as first date of diagnosis of diabetes, and the index date for controls was their matched case’s index date. c Charlson Comorbidity Index was calculated for each patient, excluding connective tissue disease. d cDMARDs included hydroxychloroquine, methotrexate, leflunomide, sulfasalazine, and tacrolimus.

**Table 2 jcm-11-02109-t002:** Risk of incident DM according to statins and corticosteroids.

	Case ^a^ (*n* = 3772)	Control ^a^ (*n* = 14,830)	Unadjusted OR (95% CI)	*p* Value	Adjusted ^b^ OR (95% CI)	*p* Value
Use of statins within a year	1466 (38.9)	3224 (21.7)	2.37 (2.19–2.56)	<0.001	2.17 (2.00–2.35)	<0.001
Cumulative corticosteroids dose within a year						
0–7 mg (1st quintile)	454 (12.0)	3283 (22.1)	1.00		1.00	
7.5–78.3 mg (2nd quintile)	604 (16.0)	3098 (20.9)	1.44 (1.26–1.64)	<0.001	1.32 (1.16–1.51)	<0.001
78.5–192 mg (3rd quintile)	716 (19.0)	3014 (20.3)	1.78 (1.56–2.02)	<0.001	1.62 (1.42–1.85)	<0.001
192.3–335 mg (4th quintile)	814 (21.6)	2901 (19.6)	2.09 (1.84–2.37)	<0.001	1.92 (1.69–2.18)	<0.001
335.5–6681.8 mg (5th quintile)	1184 (31.4)	2534 (17.1)	3.55 (3.14–4.02)	<0.001	3.20 (2.82–3.62)	<0.001

DM: diabetes mellitus, OR: odds ratio, CI: confidential interval, CCI: Charlson comorbidity index. a Data are given as number (percentage). b adjusted for CCI.

**Table 3 jcm-11-02109-t003:** Risk of incident DM according to conventional DMARDs use (≥2 prescriptions within a year).

	Case ^a^ (*n* = 3772)	Control ^a^ (*n* = 14,830)	Unadjusted OR (95% CI)	*p* Value	Adjusted ^b^ OR (95% CI)	*p* Value
Hydroxychloroquine	1497 (39.7)	6370 (43.0)	0.87 (0.80–0.93)	<0.001	0.87 (0.81–0.94)	0.001
Methotrexate	2577 (68.3)	10,210 (68.8)	0.98 (0.90–1.06)	0.594	0.84 (0.78–0.92)	<0.001
Leflunomide	1230 (32.6)	3881 (26.2)	1.38 (1.28–1.50)	<0.001	1.10 (1.02–1.20)	0.020
Sulfasalazine	695 (18.4)	2616 (17.6)	1.05 (0.96–1.16)	0.270	1.03 (0.93–1.13)	0.608
Tacrolimus	416 (11.0)	986 (6.6)	1.76 (1.56–1.99)	<0.001	1.51 (1.33–1.71)	<0.001

DM: diabetes mellitus, DMARDs: disease-modifying anti-rheumatic drugs, OR: odds ratio, CCI: Charlson comorbidity index. a Data are given as number (percentage). b Adjusted for CCI, use of steroid (cumulative prescribed dose quintiles), use of statin (yes/no).

## Data Availability

The datasets generated and/or analyzed during the current study are not publicly available due to Data Protection Laws and Regulations in Korea, but the final analyzed results are available from the corresponding authors upon reasonable request.

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
