# Peer review of "Risk of New-Onset Diabetes Mellitus Associated with Antirheumatic Drugs in Patients with Rheumatoid Arthritis: A Nationwide Population Study"

_jcm, 2022, doi:10.3390/jcm11082109_

Round 1

Reviewer 1 Report

This is an interesting cohort investigation demonstrating that some DMARDs decrease the risk of diabetes development in RA. Nevertheless some aspects need attention.

Comments 

Some information of the participants is missing (e.g. BMI, disease activity and glucose levels). This could be confounding variables and when taken into account that may impact the results. Those variables are described in discussion, but smoking is lacking. This is important in view of the etiological relationship between smoking and diabetes. Please address

Abstract: line 19: please include that the cohort exists of Korean RA patients

P 2, line 50 : please add data for HCQ

Materials and Methods:

Line 66: please add information about RA diagnosis verification

line 73. Why are those exclusioncriteria chosen? Please give a short explanation 

Line 91: tacrolimus is an uncommon drug for RA treatment, please explain

line 109. Glucocorticoid exposure was categorized only following the cumulative dosage of oral glucocorticoids. This is probably under-reported, because RA patients also use other administer forms of glucocorticoids (e.g. intra-muscular). Please discuss

Line 125: please add the incidence of RA

Line 237: "good drug compliance". This has not been studied and should be left out

PLease add a paragraph for the implications for our daily clinical practice 

Author Response

Dear Editor and reviewers,

Thank you for giving us the opportunity to revise our manuscript. We are grateful for your constructive comments, which enabled us to further improve our manuscript. We have revised the manuscript thoroughly following your comments. Our point-by-point responses to the reviewers’ comments are shown below. All revisions are highlighted in yellow in the revised manuscript.

We hope that our manuscript is now more suitable for publication in Journal of Clinical Medicine. Thank you again for your consideration.

Sincerely,

Corresponding author: Ji Seon Oh, MD, PhD

Department of Information Medicine, Big Data Research Center, Asan Medical Center, Seoul, Korea 88, Olympic-ro 43-gil, Songpa-gu, Seoul 05505, Korea

Tel.: +82-2-3010-2547; Fax: +82-2-3010-6969; E-mail: doogie55@naver.com

Corresponding author: Yong-Gil Kim, MD, PhD

Division of Rheumatology, University of Ulsan College of Medicine, Asan Medical Center

88, Olympic-ro 43-gil, Songpa-gu, Seoul 05505, Korea

Tel.: +82-2-3010-3279; Fax: +82-2-3010-6969; E-mail: bestmd2000@amc.seoul.kr

Reviewer 2 Report

I rate the article high because of the very well selected groups, both the study group and the group for comparison. This allows for the appropriate conclusions to be drawn. However, Authors should take into account the following considerations:

 1.The study shows that the treatment with DMARDs has a significant impact on the occurrence of diabetes. Therefore, in Table 1, if possible, the duration of the disease till index date should be compared. I assume that the differences emphasizing the effectiveness of prophylactic treatment of DMARDs will increase with the duration of the disease, i.e. patients poorly treated earlier will have diabetes. 

2. Closer comment in the “discussion” and “abstract” is required by the fact that DMARDs used <90 days promote the onset of diabetes, while treatment> 270 days prevents it. It is illogical to assume that the same drug causes diabetes first and then treats it. The Authors' translation "RA status" is ambiguous and requires clarification. It should be taken into account that especially classic synthetic DMARDs (to a lesser extent biological and targeted synthetic) have a delay in action of just 3 months. We see their full effect only after 6-8 months (EULAR recommendations for the management of rheumatoid arthritis with synthetic and biological disease-modifying antirheumatic drugs: 2019 update. Smolen JS, et al. Ann Rheum Dis 2020;79:685–699. doi:10.1136/annrheumdis-2019-216655). So the group of patients taking DMARDs <3 months was simply inadequately treated (in addition, we do not know whether they took the drug continuously for 3 months or only 3 months a year - in both situations it indicates improper treatment). This group improperly treated, as my own observations show, had an active disease (unfortunately the authors have no data for it) probably controlled with higher doses of corticosteroids, and these were the causes of diabetes also in the authors' material. It means that short-therm DMARDs could not cause diabetes if not caused this used chronically. I propose to remove the excerpt from < 90 days treatment from the abstract or to interpret it differently to encourage appropriate treatment.

3. I am surprised that patients in Korea are treated with tacrolimus, which does not meet EULAR and ACR standards. . (2021 American College of Rheumatology Guideline for the Treatment of Rheumatoid Arthritis. Fraenkel L et al. Arthritis Care & ResearchVol. 73, No. 7, July 2021, pp 924–939DOI 10.1002/acr.24596). It is a drug ineffective in this disease and these patients should be treated as poorly treated, where they are "made up" with corticosteroids. So he himself has nothing to do with causing diabetes. It should only be stated that in patients treated with tacrolimus diabetes occurred more often, which may encourage rheumatologists to follow global recommendations. 

4. I disagree with the statement that diabetes is caused by chronic statin use. In the present work, as in the original work cited by the Authors (Wang et al. J Am Cardiol 2012, 60, 1231-1238) there is a logical error here, not noticed by the critic of Banereje A. (Letters to the Editor: Is prolong use of statins associated with increase in the risk diabetes ?. JACC1013, 61, (9), 989). It is necessary to consider why statins were used - probably to control hyperlipidemia, which is a symptom of metabolic syndrome, which undoubtedly contributes to the onset of diabetes, the authors will also confirm this. But its root cause is obesity / overweight, the immediate cause of all of these symptoms, including the onset of diabetes. The authors of both works did not have BMI at their disposal, so they cannot claim that the firefighter who came to extinguish the fire is an arsonist. Again, it is enough to emphasize that diabetes occurred more frequently in patients using statins, probably because of the metabolic syndrome. This one, although the data in Table 1 indicate it, was completely omitted.  

5. There are no significant publications in the work on the problem of diabetes in rheumatoid arthritis that will help the Authors to interpret the results: 

Baker JF, et al. Disease activity, cytokines, chemokines and the risk of incident diabetes in rheumatoid arthritis. Ann Rheum Dis 2021;0:1–7. doi:10.1136/annrheumdis-2020-219140

Baghdadi L.: Effect of methotrexate use on the development of type 2 diabetes in rheumatoid arthritis patients: A systematic review and meta-analysis. PLOS ONE | https://doi.org/10.1371/journal.pone.0235637 July 6, 2020

Author Response

(The authors gave the same response as above.)

Round 2

Reviewer 2 Report

Currently non comment. All my comments have been taken into account. I rate the publication very highly.